

# Codon optimization, expression in *Escherichia coli*, and immunogenicity analysis of deformed wing virus (DWV) structural protein

Dongliang Fei[1,2], Yaxi Guo[1], Qiong Fan[3], Ming Li[2], Li Sun[2], Mingxiao Ma[2] and Yijing Li[1]

[1] College of Animal Medicine, Northeast Agricultural University, Haerbin, Heilongjiang, China
[2] Laboratory Animal Center, Jinzhou Normal University, Jinzhou, Liaoning, China
[3] Jinzhou Agricultural and Rural Comprehensive Service Center, Jinzhou, Liaoning, China

## ABSTRACT

**Background:** Deformed wing virus (DWV) is a serious threat to honey bees (*Apis mellifera*) and is considered a major cause of elevated losses of honey bee colonies. However, lack of information on the immunogenicity of DWV structural proteins has hindered the development of effective biocontrol drugs.

**Methods:** We optimized the *VP1*, *VP2* and *VP3* codons of DWV surface capsid protein genes on the basis of an *Escherichia coli* codon bias, and the optimized genes of *roVP1*, *roVP2* and *roVP3* were separately expressed in *E. coli* and purified. Next, the three recombinant proteins of *roVP1*, *roVP2* and *roVP3* were intramuscularly injected into BALB/c and the immunogenicity was evaluated by the levels of specific IgG and cytokines. Furthermore, anti-*roVP*-antisera (*roVP1* or *roVP2* or *roVP3*) from the immunized mice was incubated with DWV for injecting healthy white-eyed pupae for the viral challenge test, respectively.

**Results:** The optimized genes *roVP1*, *roVP2* and *roVP3* achieved the expression in *E. coli* using SDS-PAGE and Western blotting. Post-immunization, *roVP2* and *roVP3* exhibited higher immunogenicity than *roVP1* and stimulated a stronger humoral immune response in the mice, which showed that the recombinant proteins of *roVP3* and *roVP2* induced a specific immune response in the mice. In the challenge test, data regarding quantitative real-time RT-PCR (qRT-PCR) from challenged pupae showed that the level of virus copies in the recombinant protein groups was significantly lower than that of the virus-only group at 96 h post-inoculation ($P < 0.05$). Among them, the degree of neutralization using antibodies raised to the recombinant proteins are between approximately 2-fold and 4-fold and the virus copies of the *roVP3* group are the lowest in the three recombinant protein groups, which indicated that specific antibodies against recombinant proteins *roVP1*, *roVP2* and *roVP3* of DWV could neutralize DWV to reduce the virus titer in the pupae. Collectively, these results demonstrated that the surface capsid protein of DWV acted as candidates for the development of therapeutic antibodies against the virus.

Corresponding author
Yijing Li, 371751697@qq.com

## BACKGROUND

The western honey bee (*Apis mellifera*) is an important insect species for the commercial pollination of high-value crops and plays a crucial role in agricultural ecology (*Garibaldi et al., 2013*; *Škubník et al., 2017*). However, as highlighted by the elevated losses of honey bee colonies in major beekeeping regions owing to colony collapse disorder, annual declines in honey bee populations have significantly increased over the past two decades (*De Smet et al., 2012*). Multiple explanations for colony losses have been proposed, including habitat degradation and the effects of pathogens, parasites and pesticides (*Capucci & Rossi, 2006*; *Tehel et al., 2019*). Of the numerous infectious pathogens and infesting parasites, *Varroa destructor*, an ectoparasitic mite, and deformed wing virus (DWV) are regarded as serious threats to honey bee health and survival (*Budge et al., 2015*; *Natsopoulou et al., 2017*).

DWV, often considered among the most common and prevalent viruses in honey bee colonies (*Dalmon et al., 2017*; *Gisder et al., 2018*), was originally isolated from a sample of symptomatic honey bees during the early 1980s in Japan; currently, DWV is distributed worldwide, wherever *Varroa* mites are found (*Bailey & Ball, 1991*). DWV infection induces clinical symptoms that include deformed and crippled wings, discoloration and paralysis and a generally shortened lifespan (*De Miranda & Genersch, 2010*). DWV is a positive-strand RNA virus belong to the family *Iflaviridae* of the order *Picornavirales* (*Lanzi et al., 2010*). DWV is separated into three major variants: DWV-A, -B and -C (*Mordecai et al., 2016*). The International Committee on Taxonomy of Viruses recognizes two master variants of DWV-A—DWV and Kakugo virus—whereas DWV-B contains *Varroa destructor* viruses and DWV-C, which has only recently been described, is phylogenetically distinct from both A and B types (*Gisder et al., 2018*; *Mordecai et al., 2016*). DWV has a single-stranded RNA genome of approximately 10,000 nucleotides and the whole genome encodes one large, uninterrupted open-reading frame that is translated into polyproteins that are cleaved by viral proteases to produce structural and nonstructural proteins (*Bowen-Walker, Martin & Gunn, 1999*). The three major structural proteins VP1, VP2 and VP3 from one polyprotein precursor form a protomer, an elementary building block of the virus capsid (*Škubník et al., 2017*; *Koziy et al., 2019*).

Although DWV is regarded as one of the primary viruses that are associated with honeybee colony losses (*Francis, Nielsen & Kryger, 2013*), the effective medicine and method against DWV remain limited. Antibodies, as naturally occurring proteins, play an important role in neutralizing virus; thus, they have been widely utilized in the prevention and control of epidemic diseases as well as for research purposes (*Hu et al., 2019*; *Nieuwkoop, Claassens & Van der Oost, 2019*). The capability of virus neutralization and antibodies titers primarily depends on antigenic immunogenicity. It is crucial to identify an excellent immunogenic antigen for developing therapeutic antibodies. A previous study has demonstrated that viral structural proteins not only play an important role in viral infection but also possess superior immunogenicity in producing neutralizing antibodies (*Ojha, Nandani & Prajapati, 2019*; *Xie et al., 2019*; *Mahdy et al., 2019*); we thus speculate that the three major DWV proteins have possibly been

suitable targets for effective therapeutic antibody against DWV. Presently, the VP1 recombinant fragment protein of DWV has already been successfully recombinantly expressed and used for the generation of monoclonal antibodies (*Lamp et al., 2016*); however, limited studies on their antigenic differences of the three structural proteins have restricted the development of therapeutic drugs.

In the present study, we obtained the recombinant proteins *roVP1*, *roVP2* and *roVP3* of DWV in *E. coli* using bioinformatics-based codon optimization; next, BALB/c mice were immunized with the recombinant proteins and the immunogenicity of the three recombinant proteins was evaluated on the basis of the specific antibody level, lymphocyte proliferation and cytokine expression. Furthermore, the challenge test was performed to assess the virus-neutralizing ability of anti-*roVP*-antisera (*roVP1* or *roVP2* or *roVP3*). To the best of our knowledge, this is the first study to express DWV structural protein genes and evaluate their immunogenic potential.

## MATERIALS AND METHODS

### Ethics statement and animal care

All animal experiments were performed in accordance with the ethical guidelines of Jinzhou Medical University and the research protocol was approved by the Animal Care and Use Committee of Jinzhou Medical University (Approval number: 2018-012). The animal facility and rearing condition was based on laboratory animal environment and facilities (GB14925-2010) and laboratory animal feed nutrient standard (GB14924.3-2010) of China. The mice were sacrificed by carbon dioxide ($CO_2$) inhalation.

### Strains, plasmids, virus, and antibody

*Escherichia coli* DH5α and BL21 (DE3) pLysS strains were purchased from TranGen Biotech (Beijing, China). The vector pET-28a was obtained from Novagen (San Diego, CA, USA). DWV LN8/17 strain (accession number: MF770715) was isolated, identified, purified and preserved at the author's laboratory and the purification method of DWV particles was essentially described by *Lamp et al. (2016)*. 6*His-tag antibody was purchased from Proteintech Group, Inc. (Wuhan, China), and horseradish peroxidase (HRP)-labeled goat anti-mouse immunoglobin G (IgG) was obtained from Bioss ANTIBODIES (Beijing, China).

### Amplification and cloning

Using a TIAamp viral RNA kit (TIANGEN, Beijing, China), viral RNA was extracted from the purified DWV LN8/17 strain at our laboratory (MF770715). Next, first-strand complementary DNA (cDNA) was transcribed using EasyScript cDNA Synthesis SuperMix (TranGen Biotech, Beijing, China), according to the manufacturer's instructions. DWV *VP1*, *VP2* and *VP3* were amplified using polymerase chain reaction (PCR) with specific primers designed based on the nucleotide sequence of the DWV LN8/17 strain (*VP1*, forward primer: 5′-ATCGTATATTAAATTTRGCAGAGG-3′, reverse primer: 5′-GGAGAGCCAGCAGAACC-3′; *VP2*, forward primer: 5′-AACAAGG ACCTGGTAAGGTAAGTA-3′, reverse primer: 5′-TTATCCTAAAGTCACAAAAA-3′;

VP3, forward primer: 5′-CAGTGCAGGCAAAACCAGAGATG-3′, reverse primer:
5′- CGGGACAAAATGGCGAGGAG-3′). PCR was conducted as follows: initial
denaturation at 95 °C for 5 min; 25 cycles at 95 °C for 45 s and 55 °C for 40 s (*VP1*), 51 °C
for 40 s (*VP2*), or 50 °C for 40 s (*VP3*); 72 °C for 60 s; and final extension for 10 min at
72 °C. The PCR products were inserted into a pMD18-T cloning vector and sequenced
using Synbio Techology Corporation (Suzhou, China).

## Codon optimization and construction of recombinant expression vectors

*VP1*, *VP2* and *VP3* sequences of the DWV LN8/17 strain were analyzed and optimized
based on *E. coli*-preferred codons without changing the amino acid sequence of the
corresponding proteins (Gao et al., 2015; Mansouri et al., 2013; Wang et al., 2012).
High-frequency-usage codons in *E. coli* were the most commonly used for each of
the individual amino acids (Tian et al., 2017). *VP1*, *VP2* and *VP3* were subsequently
reverse-translated by applying the single-most commonly used codon for each amino
acid such that the final codon-optimized genes were represented by the most likely
nondegenerate coding sequence and online optimization software (http://www.jcat.de/
and http://genomes.urv.es/OPTIMIZER/) were utilized for codon design. The designed
genes *roVP1*, *roVP2* and *roVP3* were synthesized by Synbio Techology Corporation
(Suzhou, China) and inserted into the plasmid pET-28a(+) with the respective restriction
enzymes (*roVP1*: BamHI and HindIII; *roVP2*: EcoRI and HindIII; and *roVP3*: BamHI and
HindIII), and the 6× His tag sequence is located at the N terminus of the recombinant
plasmids.

## Protein expression and purification

For each protein, BL21 (DE3) pLysS cells were transformed with the appropriate
recombinant plasmids and plated on LB agar with 50 μg/mL kanamycin. The transformed
BL21 cells were incubated at 37 °C in an orbital shaker at 220 rpm until an $OD_{600}$ of 0.6
was obtained and then induced with 0.1 mM, 0.25 mM, or 0.5 mM isopropyl-β-D-
thiogalactoside (IPTG; Sigma–Aldrich, St. Louis, MO, USA) at 30 °C for 6 h. To verify the
expression of the target proteins, the cells were collected by centrifugation and the target
proteins were confirmed using 12% sodium dodecyl sulfate polyacrylamide gel
electrophoresis (SDS-PAGE).

For purifying the recombinant proteins, the cultures were harvested by centrifugation at
6,000 rpm for 30 min at 4 °C and the obtained cell pellets were stored at −80 °C for later
use. All recombinant proteins in inclusion bodies were purified and collected using
Ni-affinity chromatography (GE Healthsystems, Uppsala, Sweden), according to the
manufacturer's instructions. Briefly, the inclusion bodies were solubilized in 6 M urea
after being washed twice with a washing buffer (1,000 mL of 4 M urea, 1% Triton X-100,
5 mM EDTA, 5 mM β-mercaptoethanol and 50 mM Tris-HCl; pH 8.5) at room
temperature for 30 min. The purified and denatured proteins were then refolded via
dialysis in phosphate buffered saline (PBS) containing urea (initial concentration: 6 M;
decreased by 1 M/dialysis) and 0.5 M arginine. The refolded proteins were then dialyzed

five times against the dialysis buffer to eliminate arginine, changing from 0.5 M to 0 M buffers by the 12 h time-point. Recombinant *roVP1*, *roVP2* and *roVP3* proteins were confirmed using SDS-PAGE and Western blotting and stored at −80 °C until further use.

## Western blot analysis

The purified recombinant proteins were electrophoresed on 12% polyacrylamide slabs along with pre-stained protein markers on adjacent lanes and transferred to a polyvinylidene difluoride (PVDF) membrane. The membranes were blocked overnight with 3% (w/v) skim milk in Tris-buffered saline containing 0.05% Tween-20 (TBST) at 4 °C overnight, then incubated for 1 h at room temperature with an anti–His tag antibody (1:3,000) or an anti-DWV polyclonal antibody (1:500) in blocking buffer. The anti-DWV polyclonal antibody was obtained from the mice after they were injected thrice at biweekly intervals with the purified virus particles of DWV. Simultaneously, the serum of healthy mice was also used as the negative control for incubating with PVDF membrane of the recombinant proteins. After three washes, the membranes were incubated with HRP-labeled goat anti-mouse IgG (1:3,000) for 2 h at room temperature. After three 5 min washes with TBST, color development was performed with an ECL Western substrate (Thermo Scientific, Waltham, MA, USA).

## Mice immunizations

To evaluate the immunogenicity of the recombinant proteins, 4–6 week-old female BALB/c mice (five/group) were subcutaneously immunized with a mixture of 200 μL each of the purified protein (20 μg) and 200 μL of Freund's incomplete adjuvant (Sigma). All mice were then boosted twice with the same dose of Freund's incomplete adjuvant at a 2 week interval with the same recombinant proteins. Negative controls were injected with PBS and positive controls with purified virus particles. Serum samples were prepared prior to the first immunization and 2 week after each immunization and stored at −80 °C until further analysis. At 2 week after the last inoculation, the animals were sacrificed by euthanasia to isolate splenocytes for lymphocyte proliferation in response to the recombinant proteins.

## Detection of specific antibody responses using enzyme-linked immunosorbent assay

According to a previously reported method, IgG levels in mice sera were detected using ELISA (*Stachyra et al., 2016*). In brief, microtiter plates (Corning, New York, USA) were coated with 50 μL of the purified virus particles/well at a concentration of 5 μg/mL at 4 °C overnight. The plates were washed thrice with PBS containing 0.05% TBST and blocked with 2% bovine serum albumin (BSA)–PBST for 1.5 h at 37 °C. After five washes, the sera samples (1:200) were added and the plates were incubated for 1.5 h at 37 °C. After another five washes with PBST, the plates were incubated for 1 h at 37 °C with HRP-conjugated goat anti-mice IgG (1:3,000). After a final round of washes, the products were detected by adding the developing reagent tetramethylbenzidine. Absorbance was

read at a wavelength of 450 nm ($OD_{450}$) using an ELx808 Absorbance Microplate Reader (BioTek Instruments, Winooski, VT, USA).

## Lymphocyte proliferation assays and cytokine detection

Two weeks after the last immunization, spleens were harvested from five mice in each group under sterile conditions. Splenocyte suspensions were prepared and the splenocytes were cultured in 96-well plates at a density of $2.0 \times 10^6$ cells/well in Dulbecco's modified Eagle's medium containing 10% fetal bovine serum. The cells were then stimulated with the purified virus particles (10 μg/mL) and Dulbecco's modified Eagle's medium was used alone as the negative control. According to the manufacturer's instructions, the plates were incubated in 5% $CO_2$ for 24 h at 37 °C, and then a cell-counting kit 8 solution (Vazyme, Nanjing, China) was added to each well and incubated for 8 h followed by the addition of 150 μL dimethyl sulfoxide solution per well; absorbance was measured at 450 nm with a microplate reader. The stimulation index (SI) was calculated as the ratio of the $OD_{450}$ of the stimulated cells to the $OD_{450}$ of the negative controls. The cytokine levels of interleukin (IL)-2, IL-4, IL-10 and interferon gamma (IFN-γ) from serum samples were detected using a commercial ELISA kit (Dakewei, Cangzhou, China), according to the manufacturer's instructions.

## Challenge test in the honeybee pupae

For preliminarily assessing the ability of IgG antibodies from mice immunized using the recombinant proteins to neutralize DWV, we collected mice serum of the groups after the third immunization to perform the challenge test. The challenge test was modified with reference to Tehel's and Sun's methods (*Sun et al., 2018*; *Tehel et al., 2019*). Prior to the challenge test, DWV were incubated with anti-*roVP1*- antisera, anti-*roVP2*-antisera, anti-*roVP3*-antisera or anti-DWV-antisera (the mice immunized with DWV viral particles and virus purification were performed as per Lamp's description (*Lamp et al., 2016*)) for 1 h at 37 °C. A total of 30 healthy white-eyed pupae from the same colony in Wendilou Town of Jinzhou (41.25°N, 121.11°E), were randomly distributed into six groups, with each group containing five pupae. Next, the pupae were transferred to 48-well microtiter plates, and each pupa in groups 1–4 was injected with one μL of virus particles and antisera suspension containing $1 \times 10^4$ copies of DWV and 50-diluted antisera; the pupae of group 5 were only injected with a mixture of five antisera from the immunized mice and the pupae of group 6 were only injected with $1 \times 10^4$ virus particles and all the groups were maintained in incubators at 35 °C and 50% RH to monitor pupal development. At 96 h post-inoculation (p.i.), all pupae of the groups were collected, then total RNA extraction was performed according to the manufacturer's instructions (TranGen, Beijing, China) and the extracts were stored at −80 °C until further use.

The quantification of the DWV titer in the samples was performed using real time qRT-PCR using the *TransScript*® Green One-Step qRT-PCR SuperMix (TranGen, Beijing, China), and RT-qPCR reaction contained 75 ng RNA, $1 \times$ *TransScript*® Tip

Green qPCR SuperMix, 1 × Passive Reference Dye, and 5 pmol of each primer: DWV-F1 and DWV-R1 (DWV-F1: 5′-TAGTGCTGGTTTTCCTTTGTC-3′; DWV-R1: 5′-CTGTG TGAGTAATTGAATCTC-3′) with Actin F1 and R1 (Actin-F1: 5′-CTTGGAATCGCAG ATAGAATGC-3′; Actin-R1: 5′-AATTTTCATGGTGGATGGTGC-3′) as the reference gene. *Apis mellifera* β-actin, which is reported as a stable reference gene in the DWV infection, was used for normalization as described by *Chen, Higgins & Feldlaufer (2005)*. The qPCR protocol for both DWV and β-actin reference gene was as follows: 45 °C for 5 min, 95 °C for 30 s, 40 cycles of 95 °C for 5 s, 54 °C for 30 s (DWV), 55 °C for 30 s (β-actin), and 72 °C for 20 s. The relative DWV titers of the pupae were assessed by calculating delta Ct values ($\Delta Ct = Ct_{(DWV)} - Ct_{(actin)}$).

## Statistical analysis

All data are expressed as mean ± standard deviation. Statistical analyses were performed using GraphPad Prism 7.0 software (GrapPad, San Diego, CA, USA). Differences in the levels of antibodies, lymphocyte proliferation and cytokines among the various groups were evaluated using an independent samples *t*-test. $P < 0.05$ was considered statistically significant.

# RESULTS

## Codon optimization of *VP1*, *VP2* and *VP3*

After sequencing, the full lengths of *VP1*, *VP2* and *VP3* were 1248, 675 and 738 bp, encoding polypeptide of 416, 225 and 246 amino acid residues, respectively. Analysis of three structural protein genes revealed various rare codons, including CTT (Leu), CTA (Leu), CGA (Arg), AGA (Arg), AGG (Arg), GGA (Gly), ATA (Ile), CCC (Pro), TCA (Ser) and ACA (Thr) in *E. coli*. To obtain high expression level of the recombinant VP1, VP2 and VP3 proteins, *roVP1*, *roVP2* and *roVP3* were synthesized so as to replace the low-frequency-usage codons with high-frequency ones in *E. coli*. Results indicated that 71% (298/416; *roVP1*), 72% (162/225; *roVP2*) and 73% (180/246; *roVP3*) of the codons were optimized according the codon bias of the *E. coli* genome (Table S1). The sequence alignment showed that codon-optimized *roVP1*, *roVP2* and *roVP3* shared 72.8%, 72.6% and 71.4% identity with wild-type *VP1*, *VP2* and *VP3*. Furthermore, the codon adaptation index (CAI) of the three genes increased from 0.19, 0.20 and 0.19 to 0.94, 0.95 and 0.94 (Fig. S1), respectively; further, the GC content was also adjusted from 41%, 38% and 39% to 56%, 55% and 53%, respectively. The substitution of nucleotides did not alter their coding amino acid sequences.

## Expression and purification of the recombinant proteins

The recombinant plasmids containing *roVP1*, *roVP2* and *roVP3* were transformed into *E. coli* BL21 (DE3) and expressed under the induction of IPTG. The recombinant proteins roVP1 and roVP3 were expressed in a soluble form and purified from the supernatant. The recombinant protein roVP2 was expressed in an insoluble form, that is, as inclusion bodies, and was purified after solubilization in 6 M urea and by using Ni-affinity chromatography. SDS-PAGE analysis showed that the apparent molecular weights of the

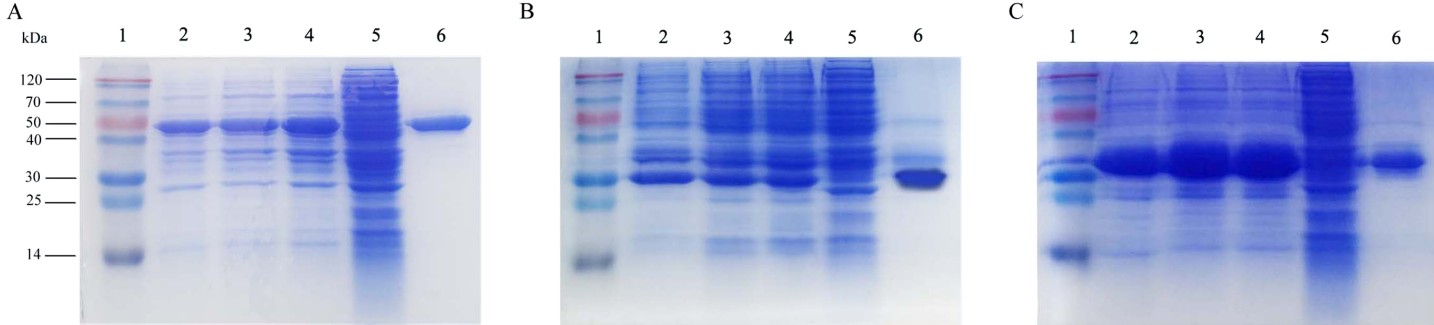

**Figure 1 Expression and purification of the recombinant proteins by SDS-PAGE analysis in *E. coli*.** (A)–(C) represent the recombinant proteins *roVP1*, *roVP2* and *roVP3*, respectively. Lane 1; protein weight marker. Lane 2–4; induced *E. coli* BL21 (DE3) of the corresponding plasmids after 6 h with 0.1 mM, 0.25 mM and 0.5 mM IPTG, respectively. Lane 5; uninduced *E. coli* BL21 (DE3) of the corresponding plasmids as control. Lane 6; purification of the corresponding recombinant proteins.                              

recombinant proteins were approximately 50.5 (*roVP1*), 29.1 (*roVP2*) and 32.6 kDa (*roVP3*), which was in accordance with their theoretical values (Fig. 1). The optimized IPTG concentrations of expression were 0.5 (*roVP1*: Fig. 1A), 0.1 (*roVP2*: Fig. 1B), and 0.25 (*roVP3*: Fig. 1C) mmol/L. After purification with Ni-affinity chromatography, the recombinant proteins were presented as a single band (*roVP1*: Fig. 1A; *roVP2*: Fig. 1B; *roVP3*: Fig. 1C).

## Western blot analysis of the recombinant proteins

To determine the expression of the recombinant proteins, Western blot analysis was performed using mouse anti–His tag antibodies and anti-DWV polyclonal antibodies, but the recombinant proteins didn't combine with the negative control (the serum of healthy mice). The results showed that the single reaction bands corresponding to the bands in SDS-PAGE could be observed in the PVDF membrane, indicating the reactogenicity of the recombinant proteins *roVP1*, *roVP2* and *roVP3* to specific antibodies (Figs. 2A–2C; Fig. S2).

## Immune effect comparison for serum antibody titers

To evaluate the immunogenicity of the recombinant proteins, BALB/c mice were immunized with *roVP1*, *roVP2* or *roVP3* thrice at 2 week intervals and the total IgG antibody levels were measured using ELISA. As shown in Fig. 3 and Table S2, after the first immunization, serum total IgG levels were higher in mice of the recombinant groups and group of the purified virus particles than in those of the control group ($P < 0.05$). In addition, the mice that were inoculated with the purified virus particles induced higher levels of IgG antibodies than those of the other groups. After the booster immunization, the antibody levels in group *roVP3* were significantly higher than those in the *roVP2* and *roVP1* groups; the *roVP1* group induced the lowest immune response. At 2 weeks after the last immunization, high levels of IgG were detected in the three recombinant protein groups compared with those in the control PBS group ($P < 0.05$). Although the difference between the *roVP3* and *roVP2* groups was insignificant ($P > 0.05$), the antibody

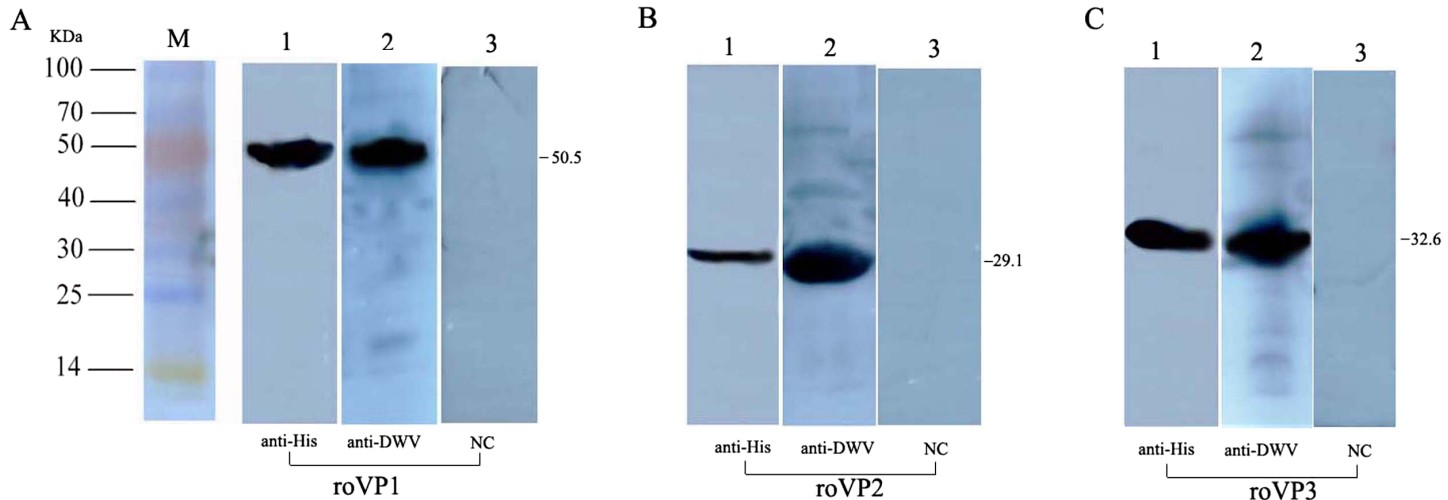

**Figure 2 Western blot identification of the recombinant proteins with the anti-His tag antibody and mouse anti-DWV polyclonal antibody.**
(A)–(C) represent *roVP1*, *roVP2* and *roVP3*, respectively. Lane 1; anti-His tag antibody was used as primary antibody. Lane 2; mouse anti-DWV polyclonal antibodies as primary antibody. Lane 3; negative control (NC), the serum of healthy mice.

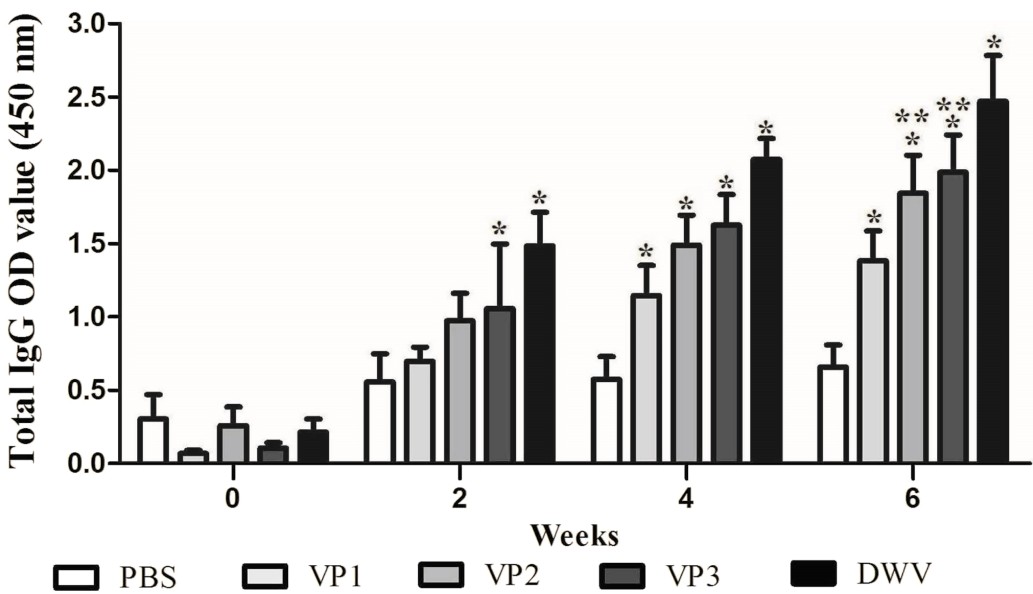

**Figure 3 Specific IgG antibodies in serum of the immunized mice.** The $OD_{450}$ value of total IgG were detected on 0, 2, 4 and 6 weeks post-vaccination. Data are expressed as the mean ± SD ($n = 5$) values. *Significantly different compared with the control group ($P < 0.05$). **Significant differences between three groups immunized with recombinant proteins ($P < 0.05$).

levels were higher in the *roVP3* group than in the *roVP2* group and both of them were significantly higher than those in the *roVP1* group ($P < 0.05$). These results suggest that the three recombinant proteins had strong immunogenicity and that roVP3 effectively promoted the antibody levels induced in mice.

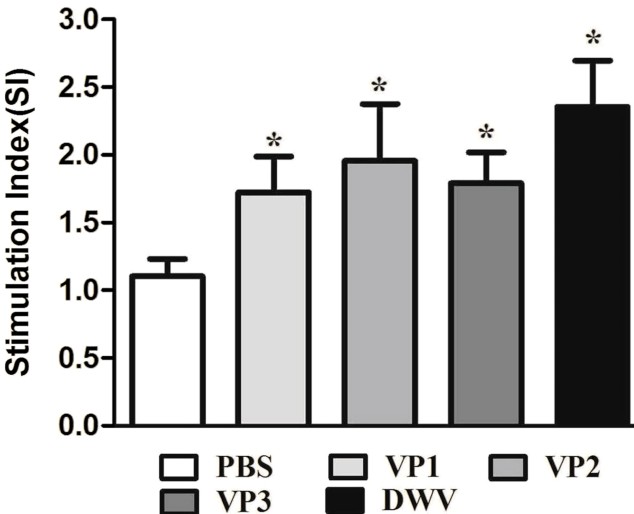

**Figure 4 Lymphocyte proliferation responses of immunized mice.** The values of the stimulation index (SI) were calculated at 6 weeks post-vaccination. Data are expressed as the mean ± SD ($n$ = 5) values. *Significantly different than the control group ($P < 0.05$).

## Immune effect comparison for lymphocyte proliferation and cytokine secretion

To evaluate splenocyte proliferation in vitro, spleen cell suspensions from the mice of different groups were prepared at 2 weeks after the last immunization. Our data suggest that a significantly higher SI value was obtained from mice immunized with the recombinant protein groups and from the group of purified virus particles than from the control groups ($P < 0.05$) (Fig. 4; Table S3). Moreover, the *roVP2* group exhibited the highest level of lymphocyte proliferation (SI = 1.96 ± 0.18), which was insignificantly different from that in the *roVP1* and *roVP3* groups; further, lymphocyte proliferation in the *roVP3* group (SI = 1.79 ± 0.10) was higher than that in the *roVP1* group (SI = 1.72 ± 0.12). These results indicate that the recombinant proteins successfully stimulated the mice to promote lymphocyte proliferation.

The levels of serum IL-2, IL-4, IL-10 and IFN-γ in each group are shown in Fig. 5 and Table S4. Compared with the PBS control group, the IL-2, IL-4, IL-10 and IFN-γ concentrations in the recombinant protein groups were significantly higher ($P < 0.05$) at 2 weeks after the last immunization. Among the recombinant protein groups, the IL-2 and IFN-γ levels were significantly higher in the *roVP2* group than in the *roVP1* and *roVP3* groups ($P < 0.05$) and the mice immunized with *roVP3* showed significantly higher levels of IL-4 ($P < 0.05$). However, the level of IL-10 was not statistically significantly different in the *roVP1*, *roVP2* and *roVP3* groups ($P > 0.05$). These results show that immunization with the recombinant proteins induced T-cells to secret multiple cytokines.

## The result of Challenge test in the honeybee pupae

The aim of challenge test was to evaluate whether antisera against DWV recombinant structural protein could neutralize virus in honeybee pupae. As seen in Fig. 6 and Table S5, the DWV levels in the group infected with only virus were significantly higher than other

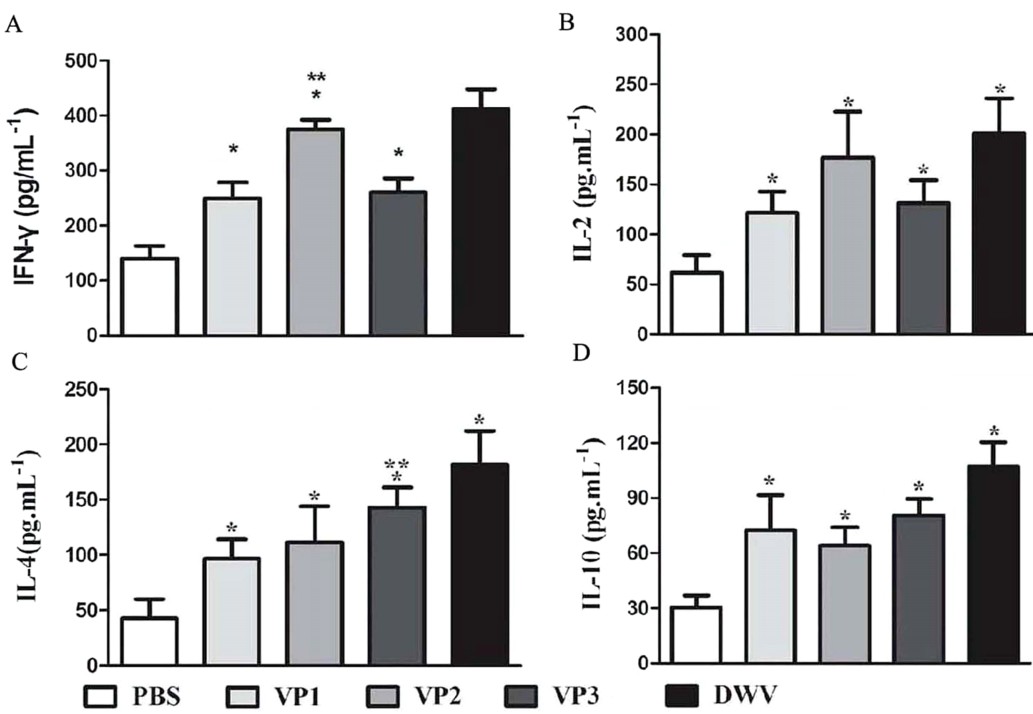

**Figure 5 Cytokine response in sera of immunized mice.** (A)–(D) represent the levels of cytokines IL-2, IL-4, IL-10 and IFN-γ by ELISA, respectively. Data are expressed as the mean ± SD ($n$ = 5) values. *Significantly different compared with the control group ($P < 0.05$). **Significant differences between three groups immunized with recombinant proteins ($P < 0.05$).

groups, indicating that DWV could massively replicate in the pupae at the 96 h. At the same time, the DWV level in the pupae of the five-antisera-only group was significantly lower than in the pupae of any other treatment groups. From the analysis of qRT-PCR data, DWV was undetectable by qRT-PCR in the group injected with five antisera. Compared with the group injected virus particles, we detected a markedly lower relative normalized virus titer in the anti-*roVP*1/2/3-antisera groups and positive control group (anti-DWV-antisera); all of them significantly differed from the virus particles group ($P < 0.05$), suggesting that anti-*roVP*1/2/3-antisera or anti-DWV-antisera could neutralize DWV virus particles. Moreover, the anti-DWV-antisera group showed the lowest virus titer in all groups (Fig. 6; Table S5). Among anti-*roVP*1/2/3-antisera groups, the relative virus titer of *roVP3* group was the lowest and it significantly differed from that in the *roVP1* groups ($P < 0.05$) (Fig. 6; Table S5). These indicated that specific antibody, which was produced in the mice immunized with the recombinant proteins, could neutralize DWV and reduce the virus titer in the pupae.

## DISCUSSION

DWV is one of the most prevalent and pathogenic viruses affecting honey bee in the world and it has been directly linked to colony collapse disorder (*Benaets et al., 2017*; *Jamnikar-Ciglenecki, Pislak Ocepek & Toplak, 2018*). At present, research of DWV is being focused on the isolation and sequence analysis of the strains, the epidemiological

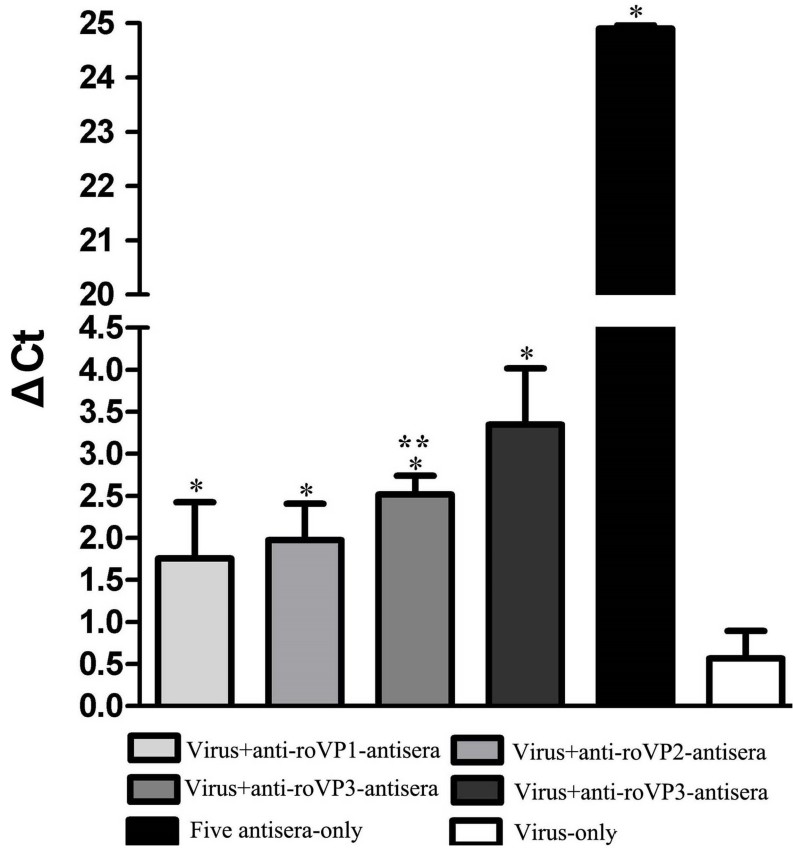

**Figure 6 Relative normalized DWV of the honeybee pupae.** The quantification of DWV titer in the samples was performed by the real time qRT-PCR, and the relative normalized DWV titer was determined as the DWV titer after normalization with the reference gene Actin and the relative DWV titers of the pupae were assessed by calculating delta Ct values ($\Delta Ct = Ct_{(DWV)} - Ct_{(actin)}$). *Significantly different compared with the virus-only group ($P < 0.05$). **Significant differences between three groups immunized with recombinant proteins ($P < 0.05$).

investigation of the interaction with *Varroa destructor*, and the establishment of a rapid detection method (*Bradford et al., 2017*; *Kevill et al., 2019*; *Tehel et al., 2019*; *Zhao et al., 2019*). However, research on therapeutic drugs against DWV is still rare. Egg yolk antibodies known as immunoglobulin Y (IgY) have been widely utilized in the prevention and control of epidemic diseases associated with *Acinetobacter baumannii*, *Vibrio splendidus*, the Dengue virus, noroviruses, rotaviruses and duck adenovirus 3 (*Fink et al., 2017*; *Ghosh, Malik & Kobayashi, 2017*; *Jahangiri et al., 2019*; *Li et al., 2016*; *Yin et al., 2019*). Moreover, *Sun et al. (2018)* reported that specific IgY could be obtained from hens immunized with an inactivated Chinese Sacbrood virus (CSBV) vaccine, which has the ability to neutralize CSBV. Therefore, we should be able to develop novel therapeutic drugs against DWV using this strategy. However, owing to lack of an efficient cell culture system and limited animal models for DWVs, we were unable to obtain large amounts of the viral protein to perform further research in this direction. At the same time, the immunogenicity and antigenic differences among the DWV structural proteins remain

unclear. In this study, we successfully produced the recombinant DWV proteins *roVP1*, *roVP2* and *roVP3* via expression in *E. coli* to evaluate their immunogenicity and Western blot analysis of the recombinant proteins conducted using anti-His monoclonal antibodies and anti-DWV polyclonal antibodies confirmed the identity of the target proteins, indicating that the DWV structural protein genes can be expressed in a prokaryotic system.

Comparing the codon usage in *E. coli*, we found many rare codons in the wild DWV structural protein genes and CAI values of the three target genes were only approximately 0.2; this likely results in lower expression yields—even expression failure—in *E. coli*. Codon optimization can improve protein production via optimization of translation elongation and previous reports have shown that a codon-optimization strategy can improve the expression level of genes in many species (*Chu et al., 2018*; *Hu et al., 2019*; *Nieuwkoop, Claassens & Van der Oost, 2019*; *Sun et al., 2017*). In this study, the rare codons of DWV *VP1*, *VP2* and *VP3* were replaced with the most highly preferred codons and the GC content was changed to approximately 70%. In addition, the AT-rich fragment was removed to avoid premature translation termination. After optimization, the CAI value of the three genes reached approximately 0.95. To avoid the impact of tag proteins on the spatial conformation of the recombinant protein, we selected pET28a (+) as the expression vector because the molecular weight of the His–tag protein is only approximately 0.8 kDa. SDS-PAGE showed that optimized *roVP1*, *roVP2* and *roVP3* were efficiently expressed in *E. coli*. Prokaryotic expression offers many advantages, such as simple manipulation, relatively low capital costs and fitting mass production (*Liu & Huang, 2018*), and serves as a convenient method for obtaining large numbers of recombinant proteins; however, protein sequences of VP1, VP2 and VP3 probably exist as potential glycosilation sites as per online software analysis (http://www.cbs.dtu.dk/services/NetNGlyc/) and the influence of potential glycosilation sites on the immunogenicity requires further research.

Research has indicated that structural proteins of the virus not only play an important role in viral invasion and replication but also have excellent immunogenicity and that they are frequently selected as candidates for the development of therapeutic antibodies (*Huo et al., 2018*; *Li et al., 2014*; *Stachyra et al., 2016*; *Wang et al., 2017*). Compared with vertebrate immune systems, honey bee immune pathways lack acquired immunity (*Bull et al., 2012*) and cannot directly generate specific antibodies in response to an antigen. We therefore hope to prepare the antibody against DWV by the mammal in the present study and humoral immunity is thought to be critical to evaluate the immunogenicity of the three recombinant proteins. Our results showed that among the three recombinant proteins, roVP3 induced the highest levels of specific IgG in the serum. IL-2 and INF-γ are mainly secreted by Th1 cells, which promote cellular immune responses and IL-4 and IL-10 are mainly secreted by Th2 cells, which promote humoral immune responses (*Wang et al., 2017*; *Zheng et al., 2019*). Comparing SI values and and cytokines, we speculate that *roVP2* has a strong effect on cellular immune responses and that *roVP3* can induce a strong humoral immune response in the mice, although all the three recombinant proteins could stimulate cellular and humoral immune responses.

In the challenge test, all the anti-roVP1/2/3-antisera could confer to the pupae protection against DWV, which indicated that antisera against recombinant protein of the mice possess the ability of neutralizing virus. Although co-injection with anti-DWV or roVP1/2/3-antisera could reduce DW levels, whether they prevent development of DWV infection need further research. In the previous research, the C-terminal extension of VP3 protein folds into a globular protruding (P) domain, exposed on the surface of the virion, which contains five Asp-His-Ser residues and the P domain may participate in receptor-binding or provide the protease, lipase, or esterase activity required for entry of the virus into a host cell (Škubník et al., 2017), and anti-roVP3-antisera show the most superior viru-neutralizing ability in challenge test; thus, we speculated that anti-roVP3-antisera inhibits virus entry into target cells by the prevention of conformational changes within P domain or the inhibition of receptor binding. These results of the challenge test provide a new idea for development of a treatment strategy against DWV.

## CONCLUSION

In summary, the present study has shown for the first time that the DWV structural proteins VP1, VP2 and VP3 can be efficiently expressed in E. coli using a codon-optimization strategy. Based on the results of mice immunization experiments and challenge test, we found that the recombinant proteins roVP3 and roVP2 have excellent immunogenicity in mice. After the virus were incubated with the specific antibodies against DWV capsid proteins, virus copies from challenged pupae reduced significantly on quantitative real-time RT-PCR detection in the challenge test, which indicated that the specific antibodies against DWV capsid proteins could neutralize DWV. Thus, they are promising candidate immunogens for the development of therapeutic antibodies.
Our research provides a novel pathway for exploiting anti-DWV biological drugs and may aid further research concerning the effects of DWV on honey bees.

### Funding
This project was supported by National Natural Science Foundation of China (31772760 and 31972626), and Liaoning Province Natural Sciences Foundation of China (20170540346). The funders had no role in study design, data collection and analysis, decision to publish, or preparation of the manuscript.

### Grant Disclosures
The following grant information was disclosed by the authors:
National Natural Science Foundation of China: 31772760 and 31972626.
Liaoning Province Natural Sciences Foundation of China: 20170540346.

### Competing Interests
The authors declare that they have no competing interests.

## Author Contributions

- Dongliang Fei conceived and designed the experiments, performed the experiments, analyzed the data, prepared figures and/or tables, and approved the final draft.
- Yaxi Guo performed the experiments, prepared figures and/or tables, and approved the final draft.
- Qiong Fan performed the experiments, analyzed the data, prepared figures and/or tables, and approved the final draft.
- Ming Li performed the experiments, authored or reviewed drafts of the paper, and approved the final draft.
- Li Sun performed the experiments, analyzed the data, authored or reviewed drafts of the paper, and approved the final draft.
- Mingxiao Ma conceived and designed the experiments, performed the experiments, authored or reviewed drafts of the paper, and approved the final draft.
- Yijing Li conceived and designed the experiments, authored or reviewed drafts of the paper, and approved the final draft.

## Animal Ethics

The following information was supplied relating to ethical approvals (i.e., approving body and any reference numbers):

The animal experiments (Name: the immunogenicity analysis of deformed wing virus (DWV) structural protein; approval number: 2018012) were carried out in accordance with the guidelines issued by the Ethical Committee of Jinzhou Medical University.

## Data Availability

The raw measurements are available in the Supplemental Files.

## Supplemental Information

Supplemental information for this article can be found online at http://dx.doi.org/10.7717/peerj.8750#supplemental-information.

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
