# Peer review of "Codon optimization, expression in Escherichia coli, and immunogenicity analysis of deformed wing virus (DWV) structural protein"

_PeerJ, doi:10.7717/peerj.8750_

## Round 0.1 · original submission · Major Revisions

Please read carefully the comments of the reviewers, and provide a revised version and a comprehensive point-by-point rebuttal letter that addresses all concerns of the reviewers.

Reviewer 1 ·

Basic reporting

The manuscript reports expressing of three major structural proteins of Deformed wing virus (DWV) in E. coli. The codons in these protein-coding nucleotide sequences were optimised to maximise expression in E. coli. The expressed recombinants VP1, VP2 and VP3 proteins ( as well as wild-type DWV particles, and PBS control) were used to immunise mice to raise antisera. The authors demonstrated that antisera reacted with the corresponding proteins in ELISA tests. Then, the authors used investigated if incubation of DWV inoculation with antisera the recombinant DWV proteins, wild-type virus particles, of control antisera could reduce infectivity of DWV in honeybee pupae injection tests. The levels DWV RNA and actin mRNA were quantified in the injected pupae by qRT-PCR. The pupae injected with DWV which were pre-incubated with antisera raised to DWV particles or any of the recombinant DWV VPs showed slightly reduced levels ( between 1 to 3 Ct values) compared to the control DWV injection ( virus only or non-specific antisera with virus). Notably, after 3 days post injection bees of all treatment groups had high levels of DWV (i.e. higher copy number of DWV genomic RNA compared to actin mRNA), including those injected with DWV particles incubated with specific antisera (Supplementary S5 Table, Fig, 6).

MS shouldpProvide more details on how antibodies developed to DWV could be applied as therapeutics agains DWV ( lines 44-45, 459-462).

The manuscript requires extensive scientific English editing. There are many spelling mistakes (e.g. line 30: "Methods" -> "Methods", line 67: "DWV is a positive, single- and plus-strand RNA virus" -> DWV is a positive-strand RNA virus").

Experimental design

Experiment summarised in Fig. 6 does not include essential virus-free control, PBS injected pupae (without virus and without any antisera). Also, it would be preferably to include injection with all five antisera (Fig. 6) without DWV.

it is important to note that, after 3 days post injection all pupae had high levels of DWV (i.e. higher copy number of DWV genomic RNA compared to actin mRNA), including those injected with DWV particles incubated with specific antisera (Supplementary S5 Table, Fig, 6). Such high levels of DWV are normally found in there bee pupae with overt DWV infection, i.e. virus-injected bees or bees exposed to Varroa mites (see for example papers where delta Ct[DWV]-Ct[actin] metrics was used - see for example https://peerj.com/articles/1591/ (Fig. 4). The reported difference between level DWV in uninfected (or covertly infected) pupae and DWV infected pupae is usually about 10 - 15 Ct units ( https://peerj.com/articles/1591/ (Fig. 4); or https://doi.org/10.1371/journal.ppat.1004230 - (Supporting Figure S1)). Therefore all treatment groups in Fig. 6 could be classified as "overtly" infected. Slight (between 1 and 3 Ct values, Supplement Table S5), though statistically significant, differences in DWV loads between the groups in Fig. 6 at 3 days post injection could be a result of different proportions of virus inoculate being inactivated by specific antibodies. Such reduction in the infectious DWV units number in the inocula is likely to result in change of virus replication dynamics, specifically increasing time when DWV infection reaches highest levels. I would suggest, therefore, to extend experiment for additional 24 for and include 96 hr time point.

Other notes.

MS does not contain protocol for purification of DWV virus particles used for immunisation (the virus particles used to raise antiserum mentioned in lines 246-247), provide method or reference.

Figure 6 legend have to include description of all treatment groups. It is not clear what "NC" means. Also note the antisera, rather than purified IgG were used. Include explanation (Materials) how "Relative normalised DWV titer" values was calculated - I understand that these were calculated as 2 in a power of Ct[DWV]-Ct[actin] (Suipplementary Table S5). The authors assumed that there is s a 2-fold difference of DWV concentration corresponding to each Ct unit, which might not be a case. it is possible just to use delta Ct values ( Ct[DWV]-Ct[actin]).

Figure 2.
roVP1 - explain why "anti-His6" and "anti DWV" antibodies stained proteins protein with different molecular weight

line 70 , Reference Lamp in not correct, use Mordecai et al 2016, - those paper proposed DWV-A-B-C.

Validity of the findings

see above

Additional comments

see above

Reviewer 2 ·

Basic reporting

The paper submitted describes an interesting basic research on the capsid proteins from DWV. Most papers on this virus are very recent, with a high impact on honey bee population. The manuscript is presented in an almost clear form, with nearly good grammar. Some sentences need to be rewritten, specially in the abstract and methodology sections.



Minor concerns:
- line 30: check "MTHODS" for missing vowel E.
- line 32-34: This sentence is not clear, please check grammar.
- line 85: speculatde
- line 108: sentence in red (font color).
- line 163: needless to say "For large scale purification", as it seems that the later expression experiment was a only a preliminary experiment.
- line 251: was injected with...
- line 247-254: experiment design is not clear: 30 pupae, 7 groups, 5 pupae/group. Were there 6 or 7 groups? What was the group 7 for?
- line 303: I suggest to replace "influence of IPTG" to "induction of IPTG".
- line 303: injected.

Experimental design

In this section, the manuscript needs more attention. I have questions on the methodology that the authors followed that are not covered in the experimental design section.

Q: Which tools were used for codon optimization?

Q: Does the authors analyze the amino acid sequence for potential glycosilation sites or other postraductional processing that may alter the recognition of viral particles?

Q: Has the production of polyclonal anti-DWV on line 185 been published before? These antibodies were produced with virus particles (like the positive controls of mice immunizations) but not the roDWV proteins.

Q: Since the virus produces a single polyprotein that should be cleaved, how do the authors known the correct amino acid sequence of each VP1, VP2 and VP3 protein?

Validity of the findings

Many vendors offer the analysis of codon optimization for gblock or gen synthesis, even for E. coli or other cell systems. So, why is this step important in the context of this research?

As a proof of concept on codon optimization for VPs production, there is no evidence showed that indicates that this strategy improved obtaining the three recombinant proteins. The paper from Lamp et al. described the recombinant production of VP1 but in low levels (DOI:10.1371/journal.pone.0164639).

Additional comments

As described in literature, the virion of DWV assemble as an immature procapsid of its VP1,VP2 and VP3, that further fills up with the viral genome. It would have been interesting to see if the humoral response or protection against DWV had improved upon a synergistic injection of roVP1, roVP2 and/or roVP3.

The production of recombinant proteins from virus or animal sources in E. coli can be a problem that the authors solved. However, the methodology and results sections described more the immunogenic potential than the recombinant expression.

Reviewer 3 ·

Basic reporting

1. The text needs English editing. The English in the present manuscript is not of publication quality and require a significant improvement. I am attaching the reviewed manuscript (in PDF) with some suggestions for modifications. However, authors should understand that the function of the reviewers is to evaluate the quality and accuracy of the research presented, and provide some feedback to improve the clarity, transparency, accuracy, and utility of potential papers, not to serve as an English spelling and grammar corrector.

2. From my perspective the literature references provide a good background and support the evidences of this study.

3. The paper is well structured, figures, and tables are fine. One reccomentadtion is to write a more attractive and clear justification of the study

Experimental design

4. Due to the lethal effects caused by the DWV, the study might be relevant and meaningful.

5. From my perspective the methodology described is appropriate and accurate., Furthermore, the methods are described in detail so that others could follow the same protocol.

Validity of the findings

6. To the best of my knowledge the literature on therapeutic strategies against the DWV is still limited. Thus, this study tested the ability of immune responses against specific targets in the virus, and may yield insights into the use of novel therapies against DWV.

7. However, the discussion section, which is the most relevant part of any study, is extremely shallow and poorly discused. In this way, authors are invited to expand the discussion. Perhaps a brief discussion on the physical blockage of the antibodies to some structures of the virus would benefit the manuscript. Authors should take some advantage of the openess of the journal to speculation, but it should be clearly stated as such.

Additional comments

From my perspective, the manuscript has the potential to contribute to solve the serious problem provoked by DWV to the populations of honey bees world wide. However,several parts of the manuscript should be rewriten so that readers are able to understand the arguments of this manuscript.

Annotated reviews are not available for download in order to protect the identity of reviewers who chose to remain anonymous.

---

## Round 0.2 · Minor Revisions

There are minor remaining issues to be addressed, please read and present a revised version attending the reviewer's concerns.

Reviewer 1 ·

Basic reporting

Revised version of MS is improved, in particular English language. Although authors did not explain how anti-DWV antisera could be directly used used as "therapeutic" agent against DWV in industrial / commercial apiculture, these antibodies will definitely be extremely valuable in DWV research. Revised version also included new interesting discussion on the potential mechanism of action of anti-DWV capsid antibodies in blocking virus entry (lines 638-652).

There are questions to the results and interpretation of "neutralisation test" experiment (lines 5127-538, Figure 6). It is clear from Supplementary Table 5 that DWV infection did develop in the pupae co-injected with the virus and with the antibodies, while in the control group of bees injected with "5 antisera mix without virus" (lines 354-365) there were no development of DWV infection at all (undetectable DWV, line 526, Supplementary Table 5). The level of DWV in the pupae of "5 antisera only" group therefore was significantly lower than in the pupae of any other treatment groups and this must be clearly stated in Fig. 6 and in Results. It should be noted that co-injection with anti-DWV antisera therefore did not prevent development of infection, but only reduced DWV levels. Maximum reduction observed when "Virus+anti-DWV-antisera" group was compared with Virus-only" group was no more than 8-fold (3 delta-Ct units, 2^3=8, assuming that one delta-Ct unit correspond to 2-fold difference of DWV RNA copy numbers). Neutralising effects in the case of antisera raised to the recombinant CPs were even lower. There were approximately 2-fold, 2.5-fold , and 4-fold reductions for anti-roVP1, anti-roVP2, and anti-roVP3 antisera, correspondingly. Although all these modest differences were statistically significant. It should be noted that in the case of the "5 antisera only group", the delta-Ct value should be at least 25 units if qPCR test had 40 cycles considering Ct(actin)=15. Therefore I suggest to specify the degree of neutralisation using antibodies raised to the recombinant proteins, approximately 2-fold to 4-fold (line 53).


Other notes :

Lines 523-526
"At the same time, the Ct value of the group with five antisera is not display using qRT-PCR, which shows that the virus was not detected in the mice antisera immunized with DWV. "
should it be replaced with:
"At the same time DWV was undetectable by qRT-PCR in the group injected with five anti-DWV antisera"...


Fig. 6. Specify in the figure legend how delta-Ct values were calculated. It is clear from the Supplementary Table 5 that these DeltaCt = Ct[DWV] - Ct[actin], but this must be specified in the main text.


line 364
Specify which five antisera were mixed ( line 364), only four antisera were listed in Fig. 6 legend.

Experimental design

see comments above

Validity of the findings

see comments above

Additional comments

none

---

## Round 0.3 · accepted · Accept

The manuscript has improved over the review rounds and it is now accepted at PeerJ.